# A Piezoelectrically Excited ZnO Nanowire Mass Sensor with Closed-Loop Detection at Room Temperature

**DOI:** 10.3390/mi13122242

**Published:** 2022-12-16

**Authors:** Xianfa Cai, Lizhong Xu

**Affiliations:** School of Mechanical Engineering, Yanshan University, Qinhuangdao 066000, China

**Keywords:** mass sensor, piezoelectric excitation, ZnO nanowires, ethanol gas, room temperature detection, closed loop detection

## Abstract

One-dimensional nanobeam mass sensors offer an unprecedented ability to measure tiny masses or even the mass of individual molecules or atoms, enabling many interesting applications in the fields of mass spectrometry and atomic physics. However, current nano-beam mass sensors suffer from poor real-time test performance and high environment requirements. This paper proposes a piezoelectrically excited ZnO nanowire (NW) mass sensor with closed-loop detection at room temperature to break this limitation. It is detected that the designed piezo-excited ZnO NW could operate at room temperature with a resonant frequency of 417.35 MHz, a quality factor of 3010, a mass sensitivity of −8.1 Hz/zg, and a resolution of 192 zg. The multi-field coupling dynamic model of ZnO NW mass sensor under piezoelectric excitation was established and solved. The nonlinear amplitude-frequency characteristic formula, frequency formula, modal function, sensitivity curve, and linear operating interval were obtained. The ZnO NW mass sensor was fabricated by a top-down method and its response to ethanol gas molecules was tested at room temperature. Experiments show that the sensor has high sensitivity, good closed-loop tracking performance, and high linearity, which provides great potential for the detection of biochemical reaction process of biological particles based on mechanics.

## 1. Introduction

The one-dimensional nanobeam resonant mass sensor has a small equivalent mass and a high resonant frequency, so its sensitivity is generally high [1,2]. It is mainly used for the measurement of small masses, and can be used to reveal the interaction force and biochemical reaction dynamics information between small masses [3], and can be used as a potential atomic or molecular scale kinetic detection [4,5,6]. As the most widespread application of nanobeam resonators, nanobeam resonant mass sensors offer many advantages such as ultra-small size, ultra-low equivalent mass, high resonant frequency, ultra-low power consumption, and large tuning range [7]. Due to these advantages, nanobeam resonant mass sensors have great potential in the field of ultra-high sensitivity mass detection. The smaller the size and the lighter the mass of the resonant structure, the higher its mass sensitivity. Using carbon nanotube resonators, frequency shifts caused by single metal atoms adsorbed on carbon nanotubes can be detected by frequency-shifted noise spectra [1,8,9,10]. One of the research hotspots for nano-resonant mass sensors is their application in the field of bioanalysis [11,12]. The resonant frequency of the nano-resonant mass sensor is very sensitive to the mass adsorbed on its surface. For example, in order to detect the pathogen biomass, the functional surface of specific antibody can be set to attract the target pathogen, and the pathogen biomass can be studied by detecting the change of the resonant frequency after the binding of antigen and antibody [13,14].

However, due to the large specific surface area of nano-beams, they are easily affected by external environment and generally need to operate in ultra-low temperature environment [15,16], while most biochemical reactions are carried out at room temperature, which limits the application of nano-beams in the reaction of biological particles. In addition, the current ultra-low temperature system still accounts for the largest proportion of the whole instrument in terms of volume, weight, and cost [17]. In order to meet the requirements of detection at room temperature, it is necessary to reduce the dissipation caused by resonant beam vibration and improve the quality factor of the sensor. Most researchers adopt micron beam sensors with large quality factors. However, due to its low sensitivity, it is difficult to reach the sensitivity required for the detection of biochemical reactions [18]. In order to improve the sensitivity and detection accuracy of the micron mass resonant sensor, researchers adopted high-sensitivity materials [19], improved detection circuit accuracy [20], ultra-low temperature and high vacuum working environment [15], and reduced damping through feedback circuit [21]. However, due to the larger equivalent mass and smaller resonant frequency of micron beam, its sensitivity is far from the level of the resonant sensor of nano-beam [16]. Researchers can only resort to detection techniques based on electronics (ion conductivity technology [22], semiconductor nanowire technology [23], atomic thin nanoribons technology [24], etc.,) or optics (optical microresonator technology [25], local plasma-resonance technology [26], plasma-enhanced spectroscopy technology [27], etc.) to obtain the information of biochemical reactions. But this requires biological and chemical reactions to produce electrical or optical effects, which greatly limits its range of applications.

In this paper, a top-down fabrication method was used to obtain ZnO NW resonators. The top-down fabrication method can adopt the resonant beam with fewer defects made by mature technology, thus reducing the large inherent loss, and can reduce the relatively large clamping loss by increasing the clamping thickness [28]. At the same time, piezoelectric excitation is adopted in this paper, which can further improve the quality factor of the sensor compared with electromagnetic excitation and electrostatic excitation [29]. Through the test, the fabricated ZnO NW resonant sensor could operate at room temperature at UHF resonant frequency, avoid ultra-low temperature equipment, and greatly expand the application of nano-beam in the field of biochemical reaction testing. A dynamic model of the multi-field coupling piezoelectric excitation ZnO NW sensor was established, and the linear operating range and sensitivity curve of the sensor were obtained. In order to improve the test accuracy, the sensitive region was set at the midpoint of the resonant beam during the fabrication of ZnO NW sensor, and the step difference of the adsorption response of a single molecule was unified. In addition, a closed-loop detection system based on lock-in amplifier was used to realize real-time detection. The experimental results showed that the designed sensor worked well at room temperature, had high sensitivity, and could well respond to ethanol gas.

## 2. Dynamic Model of the ZnO NW Sensor

### 2.1. The Forced Vibration Model

As shown in Figure 1, the midpoint of the section at the left fixed end of the resonant beam is taken as the origin of coordinates, the initial length direction of the resonant beam is taken as the ***X***-axis, and the direction perpendicular to the base is taken as the ***W***-axis. The length of the resonant part of ZnO NW is *l*, and the distance between the ***X***-axis and the gate is *h*. The distance of the sensitive layer from axis ***W*** is *d* and the mass is *m*_p_. A bias voltage *U*_g_ is applied to the gate to give the beam an initial deformation. Alternating excitation voltage $U_{sd}^{AC}=u_{e}\cos\omega t=u_{e}\cos2\pi ft$ is applied across the source and drain. Due to the inverse voltage effect, Zno NW performs alternating telescopic movements in the direction of length, and Zno NW has a axial force *P* in the length direction. Due to the restriction of the clamping end, the telescopic motion is transformed into an up and down motion perpendicular to the gate [30,31]. The unit force on ZnO NW in the lateral direction is *f_e_*(*x*,*t*) = *f_A_* + *f_V_*, where, *f_A_* is the alternating transverse force of the transformation of the axial force, and *f_V_* is the van der Waals force of the gate on ZnO NW.

### 2.2. Differential Equation of Forced Vibration

It is effective to use Euler–Bernoulli beam model to describe the resonance characteristics of one-dimensional nanoresonant beams including carbon nanotubes and ZnO NWs [32,33,34]. Under multiple physical fields, the differential formula of transverse forced vibration of the ZnO NW under piezoelectric excitation considering damping force and axial force during vibration can be expressed as follows [35]:(1)$$EI\frac{\partial^{4}\Delta w\left(x,t\right)}{\partial x^{4}}+\rho A\frac{\partial^{2}\Delta w\left(x,t\right)}{\partial t^{2}}-P_{N}\frac{\partial^{2}\Delta w\left(x,t\right)}{\partial x^{2}}+c\frac{\partial \Delta w\left(x,t\right)}{\partial t}=f_{e}\left(x,t\right)$$
where *E* is the elastic modulus of the ZnO NW, *I* is the inertia moment of the ZnO NW cross-section, *ρ* is the density of the ZnO NW, *A* is the cross-sectional area of the ZnO NW, *P_N_* is the static axial force caused by the bias voltage, *c* is the damping coefficient, corresponding to all energy dissipation during vibration.

The ZnO NW extends due to the inverse piezoelectric effect, and the theoretical elongation is $d^{T}\left(U_{sd}^{AC}+U_{g}\right)$, where $d^{T}$ is the inverse piezoelectric coefficient of ZnO NWs, which can be taken to be 20 × 10^−12^ m/V [31,36]. However, due to the limitation of fixed support at both ends, the actual deformation is [35]:(2)$$\Delta l=\frac{1}{2}\int_{0}^{l}{\left(\frac{\partial \Delta w}{\partial x}\right)}^{2}dx=\frac{1}{2}\Delta w{\left(\frac{l}{2},t\right)}^{2}\int_{0}^{l}{\left(\phi^{\prime }\right)}^{2}dx$$
where ϕ is the modal function.

According to the relevant knowledge in elastic mechanics, the axial force on the ZnO NW is:(3)$$P=-\frac{EA}{l}\left(d^{T}\left(U_{sd}^{AC}+U_{g}\right)-\Delta l\right)$$

Because Δl is much smaller than the theoretical elongation $d^{T}\left(U_{sd}^{AC}+U_{g}\right)$, the influence of Δl on *P* can be ignored. Then, the alternating axial force $P_{A}=-EAd^{T}U_{sd}^{AC}/l$ and the static axial force $P_{N}=-EAd^{T}U_{g}/l$ are contained in *P*. The alternating axial force on the ZnO NW is converted into the equivalent transverse distribution force [37]:(4)$$f_{A}\left(x,t\right)=P_{A}\frac{\partial^{2}\Delta w}{\partial x^{2}}=P_{A}\Delta w\phi^{\prime \prime }$$

Due to the small spacing between the ZnO NW and the gate, the ZnO NW is subjected to van der Waals forces exerted on it by the gate during vibration. According to reference [35], the van der Waals force *f_v_* of the gate pole on ZnO NW is:(5)$$f_{v}=f_{v1}\Delta w+f_{v2}\Delta w^{2}+f_{v3}\Delta w^{3}$$
where, $f_{v1}=\frac{H_{v}\sqrt{2r}}{8{\left(h-\bar{w_{0}}-r\right)}^{3}}$, $f_{v2}=\frac{3H_{v}\sqrt{2r}}{16{\left(h-\bar{w_{0}}-r\right)}^{4}}$, $f_{v3}=\frac{H_{v}\sqrt{2r}}{8{\left(h-\bar{w_{0}}-r\right)}^{5}}$, *r* is the radius of the ZnO NW, *H_v_* is the Hamaker constant, *H_v_* is often taken as 5.8 × 10^−19^ J, and $\bar{w_{0}}$ is the static bias average displacement of the ZnO NW.

### 2.3. Frequency Formula and Nonlinear Amplitude-Frequency Characteristic Formula

The separation of variables method is used. Let Δw(x,t)=ϕ(x)q(t), Formula (1) can be converted into:(6)$$\frac{\ddot{q}}{q}+\frac{c}{\rho A}\frac{\dot{q}}{q}-\frac{P_{A}\left|\Delta w\right|\bar{\phi^{\prime \prime }}}{\rho A\bar{\phi }q}-\frac{f_{v2}\bar{\phi }q}{\rho A}-\frac{f_{v3}{\bar{\phi }}^{2}q^{2}}{\rho A}=-\frac{EI\phi^{\left(4\right)}}{\rho A\phi }+\frac{N}{\rho A}\frac{\phi^{\left(2\right)}}{\phi }+\frac{f_{v1}}{\rho A}$$
where |Δw| is the amplitude at the midpoint position of the ZnO NW, $\bar{\phi }=\frac{1}{l}\int_{0}^{l}\left|\phi (x)\right|dx$, ${\bar{\phi }}^{2}=\frac{1}{l}\int_{0}^{l}{\left|\phi (x)\right|}^{2}dx$.

Let the left and right sides of Formula (6) equal to $-\omega_{0}{}^{2}$, $\omega_{0}$ is the resonant frequency of the ZnO NW, and let the small parameter $\varepsilon =\bar{w_{0}}/h$, then the left side of Formula (6) is the nonlinear main vibration formula:(7)$$\ddot{q}+\omega_{0}{}^{2}q+\varepsilon A_{1}\dot{q}-\varepsilon A_{2}q^{2}-\varepsilon A_{3}q^{3}-\varepsilon A_{4}\cos\omega t=0$$
where $A_{1}=\frac{ch}{\rho A\bar{w_{0}}}$, $A_{2}=\frac{hf_{v2}\bar{\phi }}{\bar{w_{0}}\rho A}$, $A_{3}=\frac{hf_{v3}{\bar{\phi }}^{2}}{\bar{w_{0}}\rho A}$, $A_{4}=\frac{-EAd^{T}u_{e}\left|\Delta w\right|\bar{\phi^{\prime \prime }}h}{\rho A\bar{\phi }l\bar{w_{0}}}$.

Using multi-scale method, the amplitude-frequency characteristic formula of the ZnO NW vibration can be obtained as follows:(8)$$\omega =\omega_{0}-\frac{3\varepsilon }{8\omega_{0}}A_{3}{\left|\Delta w\right|}^{2}\pm \varepsilon \sqrt{\frac{A_{4}{}^{2}}{4\omega_{0}{}^{2}{\left|\Delta w\right|}^{2}}-{\left(\frac{A_{1}}{2}\right)}^{2}}$$

The right side of Formula (6) is the frequency solution formula of the ZnO NW vibration:(9)$$\phi^{\left(4\right)}\left(x\right)-\alpha^{2}\phi^{\left(2\right)}\left(x\right)-\beta^{4}\phi \left(x\right)=0$$
where $\alpha^{2}=-P_{N}/EI$, $\beta^{4}=\left(\rho A\omega_{0}{}^{2}+f_{v1}\right)/EI$.

According to reference [35], the frequency formula of the ZnO NW sensor is as follows:(10)$$\left|G\right|=\left|\begin{array}{llllllll} 1 & 0 & 1 & 0 & 0 & 0 & 0 & 0\\ 0 & 0 & 0 & 0 & a_{9} & a_{8} & a_{6} & a_{7}\\ 0 & \lambda_{2} & 0 & \lambda_{1} & 0 & 0 & 0 & 0\\ 0 & 0 & 0 & 0 & \lambda_{2}a_{8} & \lambda_{2}a_{9} & -\lambda_{1}a_{7} & \lambda_{1}a_{6}\\ a_{4} & a_{5} & a_{3} & a_{1} & -a_{4} & -a_{5} & -a_{3} & -a_{1}\\ \lambda_{2}a_{5} & \lambda_{2}a_{4} & -\lambda_{1}a_{1} & \lambda_{1}a_{3} & -\lambda_{2}a_{5} & -\lambda_{2}a_{4} & \lambda_{1}a_{1} & -\lambda_{1}a_{3}\\ \lambda_{2}{}^{2}a_{4} & \lambda_{2}{}^{2}a_{5} & -\lambda_{1}{}^{2}a_{3} & -\lambda_{1}{}^{2}a_{1} & -\lambda_{2}{}^{2}a_{4} & -\lambda_{2}{}^{2}a_{5} & \lambda_{1}{}^{2}a_{3} & \lambda_{1}{}^{2}a_{1}\\ \begin{array}{l} \lambda_{2}{}^{3}a_{5}\\ +a_{10}a_{4} \end{array} & \begin{array}{l} \lambda_{2}{}^{3}a_{4}\\ +a_{10}a_{5} \end{array} & \begin{array}{l} \lambda_{1}{}^{3}a_{1}\\ +a_{10}a_{3} \end{array} & \begin{array}{l} -\lambda_{1}{}^{3}a_{3}\\ +a_{10}a_{1} \end{array} & -\lambda_{2}{}^{3}a_{5} & -\lambda_{2}{}^{3}a_{4} & -\lambda_{1}{}^{3}a_{1} & \lambda_{1}{}^{3}a_{3} \end{array}\right|=0$$
where $\lambda_{1}=\sqrt{\sqrt{\alpha_{2}{}^{4}/4+\beta_{2}{}^{4}}-\alpha_{2}{}^{2}/2}$, $\lambda_{2}=\sqrt{\sqrt{\alpha_{2}{}^{4}/4+\beta_{2}{}^{4}}+\alpha_{2}{}^{2}/2}$ $a_{1}=\sin\left(\lambda_{1}d\right)$, $a_{2}=\sin\left(\lambda_{2}d\right)$, $a_{3}=\cos\left(\lambda_{1}d\right)$, $a_{4}=\mathrm{ch}\left(\lambda_{2}d\right)$, $a_{5}=\mathrm{sh}\left(\lambda_{2}d\right)$, $a_{6}=\cos\left(\lambda_{1}l\right)$, $a_{7}=\sin\left(\lambda_{1}l\right)$, $a_{8}=\mathrm{sh}\left(\lambda_{2}l\right)$, $a_{9}=\mathrm{ch}\left(\lambda_{2}l\right)$, $a_{10}=m_{p}\omega_{{}_{0}}{}^{2}/\left(EI\right)$.

The modal function of the ZnO NW sensor is as follows:(11)$$\phi (x)=\{\begin{array}{c} \phi_{1}(x)=C_{1}\mathrm{ch}\left(\lambda_{2}x\right)+C_{2}\mathrm{sh}\left(\lambda_{2}x\right)+C_{3}\cos\left(\lambda_{1}x\right)+C_{4}\sin\left(\lambda_{1}x\right)\\ \phi_{2}(x)=C_{5}\mathrm{ch}\left(\lambda_{2}x\right)+C_{6}\mathrm{sh}\left(\lambda_{2}x\right)+C_{7}\cos\left(\lambda_{1}x\right)+C_{8}\sin\left(\lambda_{1}x\right) \end{array}\begin{array}{c} \left(0\leq x\leq d\right)\\ \left(d\leq x\leq l\right) \end{array}$$
where the constants *C*_1_, *C*_2_, *C*_3_, *C*_4_, *C*_5_, *C*_6_, *C*_7_, and *C*_8_ are obtained by the formula [G][C]=[0], and $\left[C\right]={\left[\begin{array}{cccccccc} C_{1} & C_{2} & C_{3} & C_{4} & C_{5} & C_{6} & C_{7} & C_{8} \end{array}\right]}^{T}$.

## 3. Design and Fabrication of the ZnO NW Sensor

### 3.1. Fabrication of the ZnO NW Sensor

Piezoelectric materials have been widely used in the nanogenerators [38,39,40,41], but considering the mechanical properties of piezoelectric materials and the compatibility of integrated circuits, quartz resonators and zinc oxide resonators are the main micro-nano resonators excited by piezoelectricity. For one-dimensional piezoelectrically excited nanobeam resonators, ZnO nanowires with proven fabrication processes are generally used. The ZnO nanowires used in this paper have a radius of approximately 50 nm, are single-crystalline, have a hexagonal cross-section (with piezoelectric and inverse piezoelectric properties), and have no twins or defects. Based on preliminary calculations and relevant references [31,42], the design parameters for the sensors are shown in Table 1.

In this paper, the ZnO NW sensor was fabricated by a top-down approach [42], and the fabrication process is shown in Figure 2. The substrate was made of a 2 μm uniformly oxidized silicon wafer (Figure 2a), on which a layer of PMMA photoresist with a thickness of approximately 100 nm was applied and pre-baked (temperature 394 K, time 20 min). The ZnO nanowires were then dispersed on the PMMA photoresist and a layer of PMMA photoresist with a thickness of approximately 400 nm was applied on top, a thickness that would ensure a good completion of the subsequent lift-off process was sufficient (Figure 2b) [43,44]. After the pre-baking process, both ends of the nanowires were exposed graphically by electron beam (Figure 2c). Then the copper with a thickness of 400 nm to 800 nm was then vapor deposited at a rate of 0.4 Å/s. The PMMA photoresist was then dissolved with acetone using a lift-off process to obtain a nanowire resonator with both ends clamped together. Negative adhesive was applied to the upper surface of the electrode and exposed to electron beam patterning to form a protective film on the surface of the electrode to reduce oxidation of the electrode and to reduce the noise generated by charge interference of the electrode by air molecules (Figure 2d).

Since the sensor designed in this paper needs to test its response to ethanol molecules, copper phthalocyanine (CuPc), which is sensitive to ethanol molecules, was selected as the sensitive layer. In this paper, a lift-off process was used to locate the deposited sensitive layer. The ZnO NW resonator surface was first spin-coated with a layer of PMMA photoresist of approximately 200 nm, then the part of the ZnO NW where the sensitive layer was to be deposited was spot-exposed in a scanning electron microscope (Figure 2e), followed by vapor deposition of the CuPc sensitive layer, and the ZnO NW sensor with the spot-coated sensitive layer was obtained using the lift-off process. To facilitate wiring and protect the sensor, it was finally TO-encapsulated (Figure 2f).

### 3.2. Closed-Loop Detection of the ZnO NW Sensor

In order to realize real-time detection of resonant frequency, a closed-loop control strategy combining phase-locked loop (PLL) and frequency modulation (FM) technologies was used [42]. The closed-loop detection circuit of the ZnO NW sensor is shown in Figure 3. In the figure, PS is the power divider and BPF is the bandpass filter. The signal generator (VCO) applies a piezoelectric excitation $U_{sd}^{AC}$ to the ZnO NW sensor with an excitation frequency close to the intrinsic frequency *ω*_0_ obtained from the open-loop sweep. The resonant signal from the ZnO NW is fed through a low noise amplifier (LNA) to a mixer, where it is mixed with a mixed signal containing the original excitation frequency *ω* and the low frequency reference signal Δ*ω*, resulting in a low frequency signal to be measured containing information about the intrinsic frequency of the ZnO NW sensor. The frequency *f*_Lock_ of this signal to be measured is detected by a lock-in amplifier to obtain information about the vibration of the transducer, and the VCO is tuned so that the excitation frequency can be tracked to the resonance frequency. The resonant frequency of the ZnO NW sensor is recorded by the spectrum analyzer in real time, and the data are compensated by referring to the temperature information obtained by the temperature sensor, so that the adsorption of ethanol molecules can be accurately obtained.

FM range was set to 100 kHz, ∆*ω* was set to 90 kHz, and the frequency modulation voltage Mod obtained by the lock-in amplifier combined with the feedback regulation module was set to:(12)$$\mathrm{Mod}=\left(1\times 10^{-5}\times f_{\mathrm{LOCK}}-0.9\right)\left(V\right)$$

In order to reduce frequency fluctuation, all the tests in this paper were carried out in a vacuum chamber with a vacuum degree of 1 × 10^−3^ Pa. In order to reduce the influence of temperature on the resonant frequency, the vacuum cavity was insulated by water circulation, and the temperature value was measured by thermometer in real time. The surface temperature of the sensor could be maintained at 21.0 °C to 21.2 °C and no frequency shifts due to temperature fluctuations were observed. Through the open-loop test, the linear resonant frequency of the ZnO NW sensor obtained was 417.35 MHz (quality factor 3010), while the theoretical calculated value was 433.21 MHz, with an error of only 3.8%. Considering the fabrication error, measurement error of size, residual stress generated by fabrication, etc., the ZnO NW sensor dynamic model was accurate [45,46], and the calculation error can be eliminated by further refining the measurement data of size. Through the experiment test, the maximum excitation voltage amplitude at which the vibration could be stably hysteretic was approximately 1.9 V. By adjusting the damping *c* in the theoretical model, the nonlinear degree of the theoretical amplitude-frequency characteristic curve can be adjusted to fit the experimental data, and the accurate damping of the actual experiment can be obtained.

## 4. Response of the ZnO NW Sensor to Ethanol Gas

### 4.1. Sensitivity and Working Range

From Formula (10), the curve of the variation of the resonant frequency Δ*f*_0_ with the mass of the adsorbed particles Δ*m*_p_ is obtained (Figure 4). As can be seen from the figure, with the increase of Δ*m*_p_, the variation of resonant frequency gradually increases, and the slope of the change gradually decreases. That is, with the adsorption of particles, the sensitivity of the sensor also gradually decreases. When the particles are adsorbed in the middle position of the ZnO NW, the resonant frequency changes the most. Therefore, in order to obtain greater sensitivity, the sensitive layer in this paper was coated in the middle position of the ZnO NW, where *S* = −8.1 Hz/zg.

It can be obtained from Figure 4 that when Δ*m*_p_ is less than 4 ag, the slope of the resonant frequency as a function of the adsorbed mass is approximated by a straight line, that is, the sensitivity is constant. This facilitates the calculation of biochemical reaction tests for proteins and viruses with mass changes within 4 ag.

In order to improve the signal-to-noise ratio of the sensor and facilitate closed-loop testing, the excitation voltage of the nano-beam resonator is generally set to be large, which is easy to cause nonlinear effect of the ZnO NW sensor [47]. With the adsorption of particles, the nonlinear effect of ZnO NW becomes more obvious, which makes the ZnO NW sensor produce the phenomenon of frequency shift and noise amplification, which is not conducive to the accurate acquisition of particle mass and the formation of closed loop. Therefore, before particle testing, the *u_e_* working interval should be accurately determined by further calculation according to the mass range of adsorbent particles.

According to Formulas (8) and (10), the influence of Δ*m*_p_ on amplitude-frequency characteristic curve is given when the excitation voltage amplitude *u_e_* = 1.9 V (Figure 5a). When the sensor does not adsorb particles, *u_e_* = 1.9 V just makes the vibration of the ZnO NW in a stable hysteretic state. In this state, there is no jumping phenomenon, the nonlinear effect noise is small, and the sensor is at the critical point of normal operation. However, when ZnO NW adsorbs particles, its vibration has an obvious nonlinear effect. In order to keep the sensor working properly, *u_e_* can be reduced. The maximum excitation voltage amplitude that can ensure linear vibration of the ZnO NW is set as *u*_d_, and the maximum excitation voltage amplitude that can ensure stable hysteresis is set as *u*_c_. The influence of Δ*m*_p_ on the two critical voltages can be obtained by drawing the amplitude-frequency characteristic curve (Figure 5b). It can be concluded that the critical driving voltage decreases with the increase of Δ*m*_p_. Therefore, in order to ensure the stable operation of the sensor, a reasonable driving voltage amplitude should be set according to the mass and quantity of adsorbed particles.

Since the experiment designed in this paper was mainly to verify the performance of the sensor, the maximum mass of the adsorbent particle was expected to be less than 5 ag, and in order to ensure the precision of the resonant frequency test, *u_e_* was set at 0.8 V.

### 4.2. Response of the ZnO NW Sensor to Ethanol Gas

The detection of ethanol gas by the ZnO NW sensor was carried out in the vacuum chamber, and the test device in the vacuum chamber is shown in Figure 6. The solenoid valve was used to control the on-off of ethanol gas. The gas mass flowmeter was controlled by regulating the controller D08-2E, which in turn controlled the flow of ethanol gas. The high purity ethanol gas was led to the cantilever beam by a long and thin pipe. One end of the pipe was linked to an ethanol gas reservoir, and the other end was fixed on a micro step driver to control the distance between the pipe nozzle and sensor. Since the CuPc could absorb water molecules in the gas, it was necessary to filter the water molecules as much as possible before the ethanol gas was passed into the vacuum chamber.

The main parameters affecting the performance of the sensor are quality factor, frequency fluctuation, sensitivity, and resistance to environmental interference [3]. The noise of nanorobeam vibration at room temperature is mainly thermomechanical noise, which is derived from random thermal drive in NEMS, and finally manifested as the phase noise of resonator [48]. Although limited by thermomechanical fluctuations at room temperature, frequency fluctuation can be reduced and quality factor can be increased by using advanced resonator design methods. During the test, the opening of the potentiometer of the controller was set to 5%, so that the flux rate of ethanol gas was maintained at a low value. If the distance *l*_nd_ between the nozzle and the sensor was different, the gas concentration around the sensor was also different, which affected the frequency shift speed of the resonant frequency. The effect of *l*_nd_ on the sensor response is shown in Figure 7, where the blank part of the figure indicates that the solenoid valve is off and the shaded part represents the solenoid valve is on. When the solenoid valve was in the closed position, the frequency fluctuation of the sensor was approximately 1.5 kHz. Then, based on the sensitivity of the sensor, it can be obtained that the mass resolution of the sensor is about 192 zg, which is at the advanced level (typically 0.1~300 zg) obtained for nanobeams in an ultra-low temperature environment [11,49,50] and meets the test conditions for most protein and virus biochemical reactions [11,18].

When *l*_nd_ was set to 5 mm, the average decrease rate of resonant frequency was faster in the first 6 s after gas was introduced, which was about −1250 Hz/s. After 12 s of gas introduction, the frequency shift of the resonant frequency gradually decreased, with an average speed of approximately −312 Hz/s. This may be due to the slower rise in the concentration of ethanol molecules around the resonator after 12 s. From Figure 7a, it can be obtained that CuPc adsorbed approximately 1800 zg of ethanol molecules during the 18 s time of aeration. When the solenoid valve was closed, the concentration of ethanol gas around the sensor gradually decreased and the ethanol molecules on the CuPc gradually desorbed.

When *l*_nd_ was set to 15 mm, a slight change in resonant frequency could be seen, indicating that some ethanol molecules volatilized and adsorbed onto the CuPc. The maximum offset of the resonant frequency was approximately −2.5 kHz, which corresponds to a maximum adsorption of 308 zg of ethanol molecules by the resonator. However, adsorbed ethanol molecules did not remain on the CuPc surface for long enough to desorb, as there was no stable concentration of ethanol gas around the resonator and for some periods of time there may have been almost no ethanol molecules.

When *l*_nd_ was set to 3 mm, multiple sorption-desorption cycle tests were carried out, and it was found that the adsorption and desorption rates were basically the same each time. Three of the adsorption–desorption experiments are listed, as shown in Figure 7c. The frequency shift rate of the sensor was relatively stable at approximately −1960 Hz/s and the performance of the sensor remained stable over the three replicate experiments with no degradation in performance or overall shift in device response. This was analyzed because reducing the distance between the nozzle and the sensor would increase the stability of the ethanol gas concentration around the sensor. The response and recovery rates of the sensor were essentially the same for each test, and complete desorption of ethanol molecules was achieved for multiple experiments, indicating that the sensor has good repeatability.

### 4.3. Response of the ZnO NW Sensor to Human Exhaled Ethanol Molecules

Human exhaled air is mainly composed of more than 200 kinds of organic and inorganic molecules, which contains rich information closely related to human metabolism [51,52]. Gas sensors provide a way in the field of early disease screening. For example, the detection of acetone in human exhaled air by gas sensors can meet the needs of fast and convenient monitoring and diagnosis of diabetes [53,54]. However, the proportion of trace elements exhaled by the human body is extremely low, and the content of water molecules is large, which brings great challenges to the selectivity and precision of various traditional gas analysis equipment (such as gas chromatographs [55]). In this paper, the sensor was used to detect the ethanol molecules in the exhaled breath of the human body after drinking liquor to verify the sensitivity of the sensor to ethanol molecules.

The principle of the sensor for detecting ethanol molecules exhaled by the human body is shown in Figure 8a. In order to avoid the influence of water molecules and heat transfer on the resonant frequency, the exhaled gas from the human body needs to be dehumidified and thermally desorbed so that the collected gas can be used as the gas to be measured in direct contact with the resonator.

The gas exhaled from the nose was collected and passed through the dehumidification filter and the transmission distance was increased to make the gas temperature roughly the same as the temperature of the surrounding environment. The collected gas was tested using the sensor and the test results are shown in Figure 8b, where the blank part of the graph indicates that the solenoid valve is closed and the shaded part represents that the solenoid valve is open. A clear downward trend in resonant frequency can be found, with the maximum value of the resonant frequency change being about −24 kHz, which corresponds to the adsorption of about 2960 zg of ethanol molecules by the resonator. When the ethanol gas dissipated, the resonant frequency did not return to its initial position. The reason for this is that the organic molecules in the gas adhered to the surface of the cantilever beam and were difficult to desorb. However, the resonator designed in this paper allows the resonant frequency to be fine-tuned by adjusting the bias voltage (increasing the bias voltage increases the resonant frequency [56]), which has important implications for commercial applications of the resonator.

Since the filter used in this paper did not filter all the molecules that could be adsorbed on CuPc, it was inevitable that molecules other than the ethanol molecules would also cause the resonance frequency to drop, but the concentration of ethanol molecules in the gas could still be qualitatively analyzed according to the decreasing rate of resonance frequency. Follow-up experiments can be carried out with additional layers of filters for the adsorption of specific molecules to reduce the interference of these factors with the detection of ethanol molecule concentrations. Due to the limitations of fabrication accuracy and the size of ZnO NWs used, the mass resolution of the sensor proposed and fabricated by the manuscript did not reach the molecular level, which limited the analysis of chemical reactions at the molecular scale. The smaller ZnO NWs can be used to achieve mass resolution at the molecular or even atomic level at room temperature.

## 5. Conclusions

This paper designs a highly sensitive, real-time closed-loop detection, reliable and accurate ZnO NW sensor operating at room temperature. The ZnO NW sensor with a sensitive layer coated in the middle was fabricated by a top-down method and the sensor was driven by piezoelectricity. Because of the better fabrication method and excitation method, the fabricated nanobeam sensor could work at room temperature, which improves the application scenario of the sensor. The dynamic model of the ZnO NW mass sensor with piezoelectric excitation was established, and the frequency and amplitude-frequency characteristic formulas of the sensor were obtained. According to the solution of the kinetic model and the calibration of the preliminary experiment, the influence of the adsorbed particle mass on the amplitude-frequency characteristics of the sensor and the critical excitation voltage was obtained, and the operating range of the excitation voltage corresponding to the different adsorbed particle mass was obtained. The sensitivity of the sensor is −8.1 Hz/zg, and the mass resolution of the sensor is about 192 zg, reaching the biochemical reaction test conditions of most proteins and viruses. The designed piezoelectric excitation ZnO NW mass sensor had a good response ability to ethanol molecules. By changing the sensitive layer and combining with the gas component separation instrument, the sensor can be used for the rapid and convenient detection of trace components in human exhaled gas.

## Figures and Tables

**Figure 1 micromachines-13-02242-f001:**
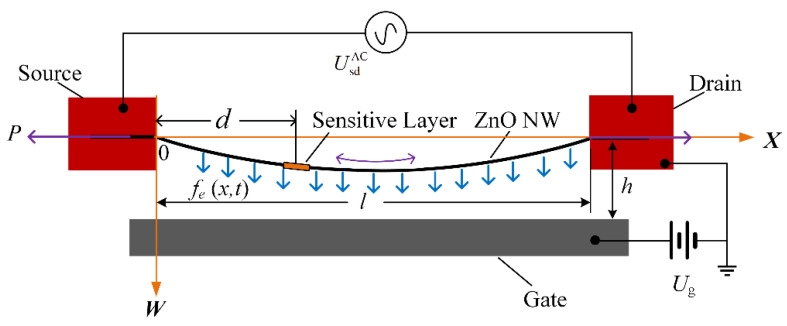
Forced vibration model of the ZnO NW sensor.

**Figure 2 micromachines-13-02242-f002:**
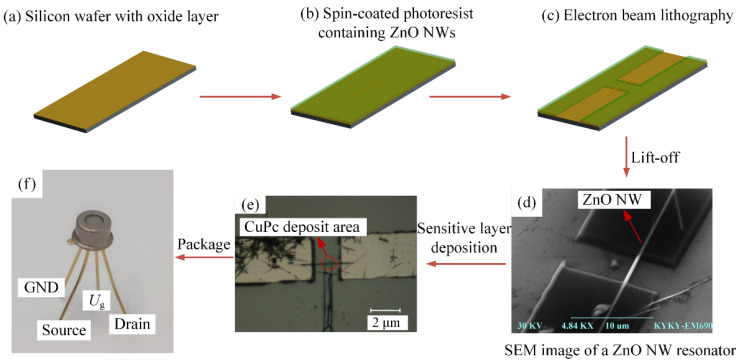
Fabrication process of the ZnO nanowire sensor.

**Figure 3 micromachines-13-02242-f003:**
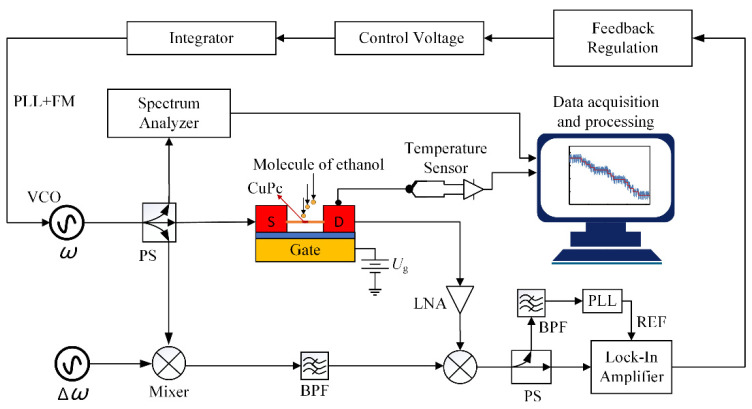
Closed loop detection circuit of the ZnO NW sensor.

**Figure 4 micromachines-13-02242-f004:**
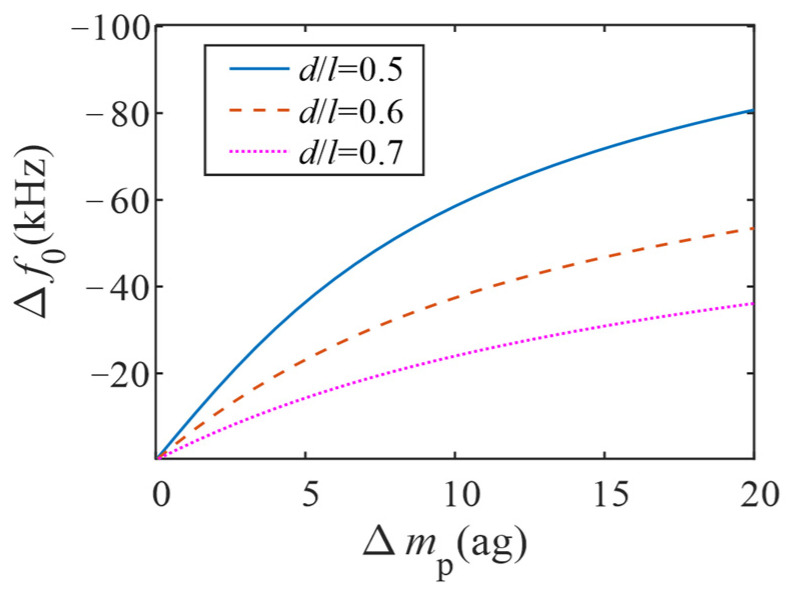
Effect of mass and position of adsorbed particles on resonant frequency.

**Figure 5 micromachines-13-02242-f005:**
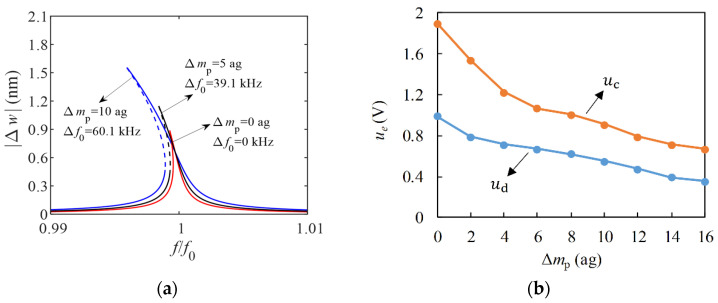
Effect of Δ*m*_p_ on resonance characteristics: (**a**) Effect of Δ*m*_p_ on the amplitude-frequency characteristic curve; (**b**) effect of Δ*m*_p_ on critical drive voltage.

**Figure 6 micromachines-13-02242-f006:**
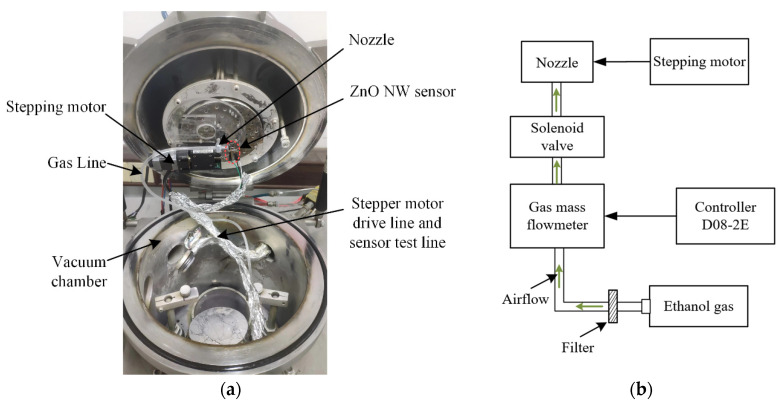
Diagram of the detection device: (**a**) Arrangement of test devices in vacuum cavities; (**b**) schematic diagram of the controlled flow of ethanol gas.

**Figure 7 micromachines-13-02242-f007:**
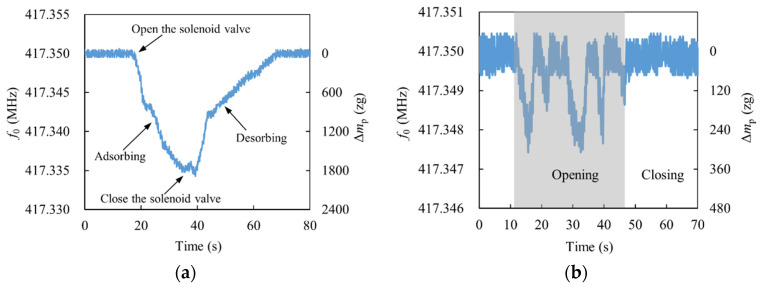
Effect of *l*_nd_ on the sensor response: (**a**) *l*_nd_ = 5 mm; (**b**) *l*_nd_ = 15 mm; (**c**) *l*_nd_ = 3 mm.

**Figure 8 micromachines-13-02242-f008:**
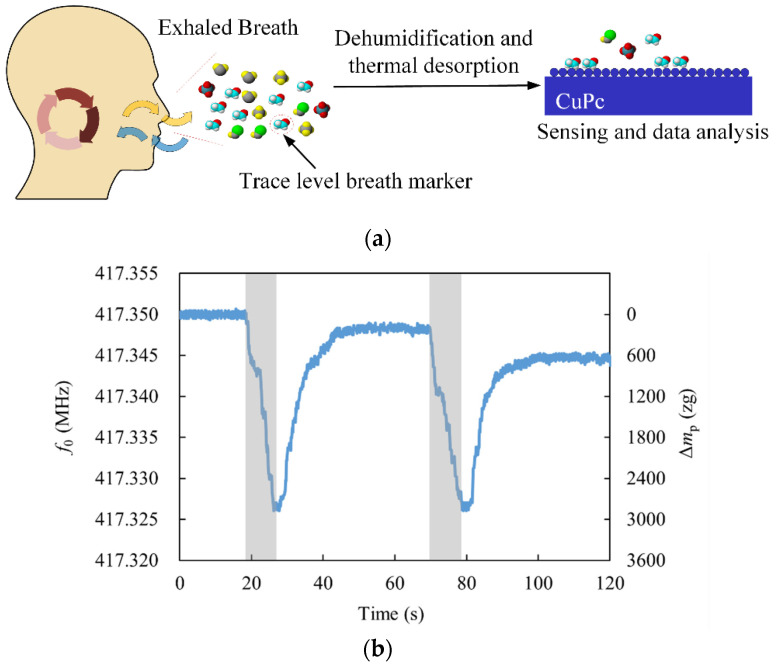
Detection of ethanol in human exhaled breath: (**a**) Schematic diagram of the sensor for detecting human exhaled air; (**b**) the response of the sensor to the detection of human exhaled gas twice in a row.

**Table 1 micromachines-13-02242-t001:** Design parameters for the ZnO NW sensor.

*l* (μm)	*r* (nm)	*h* (nm)	*m*_p_ (ag)	*U*_g_ (V)	*E* (Pa)	*ρ* (kg/m^3^)	*c* (Ns/m)
1.2	50	100	500	5	2.1 × 10^11^	5.67 × 10^3^	2.5 × 10^−11^

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
