# Peer review of "A Piezoelectrically Excited ZnO Nanowire Mass Sensor with Closed-Loop Detection at Room Temperature"

_micromachines, 2022, doi:10.3390/mi13122242_

Round 1

Reviewer 1 Report

This manuscript reports an interesting piezoelectric ZnO nanowire (NW) mas sensor with close-loop detection at room temperature. This sensor has a resonant frequency of 417.35 MHz, a quality factor of 3010, a mass sensitivity of -8.1 Hz/zg, and a resolution of 192 zg. This sensor has potential application for the detection of biochemical reaction process of biological particles. This manuscript can be improved considering the following comments:
1.-All the parameters used in the equations must be described.
2.-The description of the sub-section 2.2 must be significantly improved.
3.-The authors must check the correct reference on the inverse piezoelectric coefficient of ZnO NWs of 20x10-12 m/V.
4.-The equation (2) should include a reference. The authors must add more information on the parameters used in the equations. For instance, the authors should add the function of amplitude and modal function used in equation (2). In addition, the parameters Usd and Ug  used in the function of line 112 must be described.
5.-The authors must improve the description of all the equations by considering more information on stages and functions used to obtain these equations.
6.-The reference of equation 4 must be revised. The reference [36] does not include equation (4).
7.-The authors must revise the reference [35] used for equations (5) and (10) of ZnO NW. The reference [35] is on a resonant carbon nanotube mass sensor.
8.-The authors must improve the description of equation (5). What was the procedure to obtain equation (5)?
9.-The technical description of the different instruments of the experimental setup (Figure 6) must be improved.
10.-The authors should include discussions on the noise in the measurements of the mass sensor and the main parameters that affect the sensor performance.
11.-Which are the main challenges or limitations of the proposed sensor?

Author Response

Responses

Manuscript ID micromachines-2098811 entitled "A piezoelectrically excited ZnO Nanowire mass sensor with closed-loop detection at room temperature".

Reviewer: 1

Comments to Author

This manuscript reports an interesting piezoelectric ZnO nanowire (NW) mas sensor with close-loop detection at room temperature. This sensor has a resonant frequency of 417.35 MHz, a quality factor of 3010, a mass sensitivity of -8.1 Hz/zg, and a resolution of 192 zg. This sensor has potential application for the detection of biochemical reaction process of biological particles. This manuscript can be improved considering the following comments:

  1. 1. All the parameters used in the equations must be described.

For the comment, we give revision as below:

All the parameters in the equations have been checked and described.

  1. The description of the sub-section 2.2 must be significantly improved.

For the comment, we give revision as below:

The acquisition process for equations (2), (3) and (5) has been revised and a clerical error in equation (2) has been corrected to improve sub-section 2.2.

  1. The authors must check the correct reference on the inverse piezoelectric coefficient of ZnO NWs of 20x10-12m/V.

For the comment, we give revision as below:

The inverse piezoelectric coefficient of the ZnO NW have been double-checked by referring to the relevant references, and relevant references have been added.

[36] Lee, H.; Park, J.; Han, S.A.; Lee, D.; Kim, K.B.; Lee, N.S.; Park, J.Y.; Seo, Y.; Lee, S.; Choi, Y.J. The stress-dependent piezoelectric coefficient of ZnO wire measured by piezoresponse force microscopy. Scripta mater. 2012, 66, 101-104. https://doi.org/10.1016/j.scriptamat.2011.10.013.

  1. The equation (2) should include a reference. The authors must add more information on the parameters used in the equations. For instance, the authors should add the function of amplitude and modal function used in equation (2). In addition, the parameters Usdand Ugused in the function of line 112 must be described.

For the comment, we give revision as below:

Reference [35] has been added to equation (2). The equation (2) in the manuscript can be derived from the equations (2) and (3) in the reference [35].

The differential equation established in the manuscript contains some parameters, such as modal function, amplitude function and so on. These parameters are further solved in the sub-section 2.3. For example, the amplitude function and the modal function can be obtained by equations (8) and (11) respectively.

The parameters  and Ug used in the function on line 112 are already described in the description of figure 1 (lines 90 and 91).

  1. The authors must improve the description of all the equations by considering more information on stages and functions used to obtain these equations.

For the comment, we give revision as below:

The derivation of all the equations in the manuscript has been checked, and the calculated data of the formulas have been compared with the experimental data to ensure that the formulas are correct.

Through the open-loop test, the linear resonant frequency of the ZnO NW sensor obtained was 417.35 MHz, while the theoretical calculated value was 433.21 MHz, with an error of only 3.8%.

  1. The reference of equation 4 must be revised. The reference [36] does not include equation (4).

For the comment, we give revision as below:

The reference for equation (4) have been checked and changed. According to equation (8) in the reference [37], equation (4) in the manuscript can be derived.

[37] Devoe, D.L.J.S.; Physical, A.A. Piezoelectric thin film micromechanical beam resonators. Sensor actuat. a-phys. 2001, 88, 263-272. https://doi.org/10.1016/S0924-4247(00)00518-5.

  1. The authors must revise the reference [35] used for equations (5) and (10) of ZnO NW. The reference [35] is on a resonant carbon nanotube mass sensor.

Response:

It is effective to use Euler-Bernoulli beam model to describe the resonance characteris-tics of one-dimensional nanoresonant beams including carbon nanotubes and ZnO NWs [32-34]. The model in the manuscript is similar to the model in the reference [35], and the differential equation solving methods of the two are the same. The solution methods of modal function and frequency equation in the manuscript are derived from reference [35].

  1. The authors must improve the description of equation (5). What was the procedure to obtain equation (5)?

For the comment, we give revision as below:

The acquisition of equation (5) has been described and the reference of equation (5) has been specified.

  1. The technical description of the different instruments of the experimental setup (Figure 6) must be improved.

For the comment, we give revision as below:

Figure (6) has been described in more detail to make the description more complete.

  1. The authors should include discussions on the noise in the measurements of the mass sensor and the main parameters that affect the sensor performance.

For the comment, we give revision as below:

The main noise source of the sensor has been analyzed, and the frequency fluctuation parameter has been introduced, and the influence of temperature on this parameter has been analyzed.

The main parameters that affect the performance of the sensor have been added and analyzed.

The main parameters affecting the performance of the sensor are quality factor, frequency fluctuation, sensitivity and resistance to environmental interference [3]. The noise of nanorobeam vibration at room temperature is mainly thermomechanical noise, which is derived from random thermal drive in NEMS, and finally manifested as the phase noise of resonator [48]. Although limited by thermomechanical fluctuations at room temperature, frequency fluctuation can be reduced and quality factor can be increased by using advanced resonator design methods.

Through the open-loop test, the linear resonant frequency of the ZnO NW sensor obtained was 417.35 MHz (quality factor 3010), while the theoretical calculated value was 433.21 MHz, with an error of only 3.8%.

Therefore, in order to obtain greater sensitivity, the sensitive layer in this paper was coated in the middle position of the ZnO NW, where S=-8.1 Hz/zg.

When the solenoid valve was in the closed position, the frequency fluctuation of the sensor was approximately 1.5 kHz. Then, based on the sensitivity of the sensor, it can be obtained that the mass resolution of the sensor is about 192 zg, which is at the advanced level (typically 0.1~300 zg) obtained for nanobeams in an ultra-low temperature environment [11,49,50] and meets the test conditions for most protein and virus biochemical reactions [11,18].

The surface temperature of the sensor could be maintained at 21.0°C to 21.2°C and no frequency shifts due to temperature fluctuations were observed.

These words have been added in section 4.2.

[48] Mohanty, P.; Da., H.; Kl., E.; Yt., Y.; Mj., M.; Ml., R. Intrinsic dissipation in high-frequency micromechanical resonators Phys. rev. b, Condensed Matter 2002, 66. https://doi.org/10.1103/PhysRevB.66.085416.

  1. Which are the main challenges or limitations of the proposed sensor?

For the comment, we give revision as below:

Due to the limitations of fabrication accuracy and the size of ZnO NWs used, the mass resolution of the sensor proposed and fabricated by the manuscript did not reach the molecular level, which limited the analysis of chemical reactions at the molecular scale. The smaller ZnO NWs can be used to achieve mass resolution at the molecular or even atomic level at room temperature.

These words have been added in section 4.3.

Reviewer 2 Report

This manuscript designed a piezoelectrically excited ZnO nanowire (NW) mass sensor. This work is interesting. However, there are many problems in the manuscript. This work may be considered to be published in Micromachines after being carefully addressed. The detailed comments are as follows.

1.     There are many kinds of piezoelectric materials, why choose ZnO? What are the advantages of this article?

2.     The preparation method of ZnO NW should be described in detail. Characterization of ZnO NW including XRD and SEM should be performed comprehensively.

3.     To clarify the good repeatability of the ZnO NW sensor to ethanol gas, more adsorption-desorption cycle tests should be performed.

4.     When the ZnO NW sensor was used to detect human exhaled ethanol molecules, the resonant frequency did not return to its initial position when the ethanol gas dissipated. Therefore, the practical application of the sensor is very limited. Please solve this problem rather than just analyze the cause.

5.     The performance of the ZnO NW sensor, such as sensitivity, should be compared with published literature.

6.      Relevant articles need to be cited. 10.1007/s40820-021-00779-0; 10.1016/j.nanoen.2022.107876; 10.1016/j.nanoen.2021.105826.

Author Response

Responses

Manuscript ID micromachines-2098811 entitled "A piezoelectrically excited ZnO Nanowire mass sensor with closed-loop detection at room temperature".

Reviewer: 2

Comments to Author

This manuscript designed a piezoelectrically excited ZnO nanowire (NW) mass sensor. This work is interesting. However, there are many problems in the manuscript. This work may be considered to be published in Micromachines after being carefully addressed. The detailed comments are as follows.

  1. There are many kinds of piezoelectric materials, why choose ZnO? What are the advantages of this article?

For the comment, we give revision as below:

Piezoelectric materials have been widely used in the nanogenerators [38-41], but considering the mechanical properties of piezoelectric materials and the compatibility of integrated circuits, quartz resonators and zinc oxide resonators are the main micro-nano resonators excited by piezoelectricity. For one-dimensional nanobeam resonators, ZnO NWs with mature production process is generally used.

Two unprecedented achievements of the manuscript are as below:

(1). One-dimensional nanobeam mass sensors offer an unprecedented ability to measure tiny masses or even the mass of individual molecules or atoms, enabling many interesting applications in the fields of mass spectrometry and atomic physics. However, current nano-beam mass sensors suffer from poor real-time test performance and high test environment requirements.

In this paper, a top-down fabrication method was used to obtain ZnO NW resonators. The top-down fabrication method can adopt the resonant beam with fewer defects made by mature technology, thus reducing the large inherent loss, and can reduce the relatively large clamping loss by increasing the clamping thickness. Piezoelectric excitation is adopted in this paper, which can further improve the quality factor of the sensor compared with electromagnetic excitation and electrostatic excitation. Through the test, the fabricated ZnO NW resonant sensor could operate at room temperature at UHF resonant frequency, avoid ultra-low temperature equipment, and greatly expand the application of nano-beam in the field of biochemical reaction testing.

(2). The dynamic model of the ZnO NW mass sensor with piezoelectric excitation was established, and the frequency and amplitude-frequency characteristic formulas of the sensor were obtained. According to the solution of the kinetic model and the calibration of the preliminary experiment, the influence of the adsorbed particle mass on the amplitude-frequency characteristics of the sensor and the critical excitation voltage was obtained, and the operating range of the excitation voltage corresponding to the different adsorbed particle mass was obtained.

Some words have been added to the manuscript.

  1. The preparation method of ZnO NW should be described in detail. Characterization of ZnO NW including XRD and SEM should be performed comprehensively.

Response:

In the section on the fabrication of ZnO NW sensors, the focus of the manuscript is on the fabrication of ZnO NW sensors using the fabricated ZnO NW. The ZnO NWs used in the manuscript were purchased and the fabrication of the ZnO NWs is not the subject of this study. Therefore, the manuscript only introduces the performance of the ZnO NWs as follows.

The ZnO nanowires used in this paper have a radius of approximately 50 nm, are single-crystalline, have a hexagonal cross-section (with piezoelectric and inverse piezoelec-tric properties) and have no twins or defects.

  1. To clarify the good repeatability of the ZnO NW sensor to ethanol gas, more adsorption-desorption cycle tests should be performed.

For the comment, we give revision as below:

In the study, when lnd was set at 3 mm, multiple sorption-desorption cycle tests were carried out, and it was found that the adsorption rate and desorption rate were basically the same each time. Three of the adsorption-desorption experiments are listed, as shown in figure 7c. When adsorbed, the sensor's frequency shift velocity is relatively stable, about -1960 Hz/s.

These words have been added in section 4.2.

  1. When the ZnO NW sensor was used to detect human exhaled ethanol molecules, the resonant frequency did not return to its initial position when the ethanol gas dissipated. Therefore, the practical application of the sensor is very limited. Please solve this problem rather than just analyze the cause.

For the comment, we give revision as below:

Because the gas exhaled by the human body contains some substances that the sensor is difficult to disattach, the resonant frequency may not return to the initial position. A multilayer filter was used to filter these large particles in the experiment. To illustrate the role of the multilayer filter, special cases in multiple experiments are listed in Figure 8c. In actual applications, multi-layer filters can be set. Because the increased bias voltage can increase the resonance frequency, the resonance frequency can be restored by adjusting the bias voltage [56].

Some words have been added in section 4.3.

[56] Sazonova, V.; Yaish, Y.; Ustunel, H.; Roundy, D.; Arias, T.A.; McEuen, P.L. A tunable carbon nanotube electromechanical oscillator. Nature 2004, 431, 284-287. https://doi.org/10.1038/nature02905.

  1. The performance of the ZnO NW sensor, such as sensitivity, should be compared with published literature.

For the comment, we give revision as below:

Then, based on the sensitivity of the sensor, it can be obtained that the mass resolution of the sensor is about 192 zg, which is at the advanced level (typically 0.1~300 zg) obtained for nanobeams in an ultra-low temperature environment [11,49,50] and meets the test conditions for most protein and virus biochemical reactions [11,18].

Some words have been added in section 4.3.

  1. Relevant articles need to be cited. 10.1007/s40820-021-00779-0; 10.1016/j.nanoen.2022.107876; 10.1016/j.nanoen.2021.105826.

For the comment, we give revision as below:

Piezoelectric materials have been widely used in the nanogenerators [38-41], but considering the mechanical properties of piezoelectric materials and the compatibility of integrated circuits, quartz resonators and zinc oxide resonators are the main micro-nano resonators excited by piezoelectricity. 

[38] Zhou, L.L.; Zhu, L.P.; Yang, T.; Hou, X.M.; Du, Z.T.; Cao, S.; Wang, H.L.; Chou, K.C.; Wang, Z.L. Ultra-Stable and Durable Piezoelectric Nanogenerator with All-Weather Service Capability Based on N Doped 4H-SiC Nanohole Arrays. Nano-micro lett. 2022, 14. https://doi.org/10.1007/s40820-021-00779-0.

[39] Zhou, L.L.; Yang, T.; Fang, Z.; Zhou, J.D.; Zheng, Y.P.; Guo, C.Y.; Zhu, L.P.; Wang, E.H.; Hou, X.M.; Chou, K.C.; et al. Boosting of water splitting using the chemical energy simultaneously harvested from light, kinetic energy and electrical energy using N doped 4H-SiC nanohole arrays. Nano Energy 2022, 104. https://doi.org/10.1016/j.nanoen.2022.107876.

[40] Zhou, L.L.; Yang, T.; Zhu, L.P.; Li, W.J.; Wang, S.Z.; Hou, X.M.; Mao, X.P.; Wang, Z.L. Piezoelectric nanogenerators with high performance against harsh conditions based on tunable N doped 4H-SiC nanowire arrays. Nano Energy 2021, 83. https://doi.org/10.1016/j.nanoen.2021.105826.

[41] Wang, Y.; Zhu, L.P.; Du, C.F. Progress in Piezoelectric Nanogenerators Based on PVDF Composite Films. Micromachines 2021, 12. https://doi.org/10.3390/mi12111278.

Round 2

Reviewer 1 Report

The authors have improved their manuscript considering the reviewer's comments.

Reviewer 2 Report

Accepted.